Effects of wind speed and wind direction on crop yield forecasting using dynamic time warping and an ensembled learning model

http://orcid.org/0009-0006-2629-5208 Bediako-Kyeremeh Bright 1
http://orcid.org/0000-0003-2320-1692 Ma TingHuai 2 3 thma@nuist.edu.cn
Rong Huan 4
Osibo Benjamin Kwapong 2
Mamelona Lorenzo 1
Nti Isaac Kofi 5
Amoah Lord 2
1 School of Electronic and Information Engineering, Nanjing University of Information Science and Technology , Nanjing, Jiangsu , China
2 School of Software, Nanjing University of Information Science and Technology , Nanjing, Jiangsu , China
3 School of Computer Engineering, Jiangsu Ocean University , Lianyungang, Jiangsu , China
4 School of Artificial Intelligence, Nanjing University of Information Science and Technology , Nanjing, Jiangsu , China
5 Department of Information Technology, University of Cincinnati , Cincinnati, Ohio , USA
Mahmood Haider
Electronic publication date: 2024 Jun 11
Publication date: 2024
Volume: 12
Electronic Location ID: e16538
Received 2023 Jul 20; Accepted 2024 Apr 29
Copyright: © 2024 Bediako-Kyeremeh et al.
Copyright year: 2024
Copyright holder: Bediako-Kyeremeh et al.
License: This is an open access article distributed under the terms of the Creative Commons Attribution License, which permits unrestricted use, distribution, reproduction and adaptation in any medium and for any purpose provided that it is properly attributed. For attribution, the original author(s), title, publication source (PeerJ) and either DOI or URL of the article must be cited.
License URL: https://creativecommons.org/licenses/by/4.0/

Keywords: DTW, Wind speed, Wind direction, Ensembled, LSTM, Cashew, Crop yield, Sustainable farming

Funding: National Natural Science Foundation of China 62372243 and 6210187 This work is supported by the National Natural Science Foundation of China (No. 62372243 and No. 6210187). There was no additional external funding received for this study. The funders had no role in study design, data collection and analysis, decision to publish, or preparation of the manuscript.

==============================
The cultivation of cashew crops carries numerous economic advantages, and countries worldwide that produce this crop face a high demand. The effects of wind speed and wind direction on crop yield prediction using proficient deep learning algorithms are less emphasized or researched. We propose a combination of advanced deep learning techniques, specifically focusing on long short-term memory (LSTM) and random forest models. We intend to enhance this ensemble model using dynamic time warping (DTW) to assess the spatiotemporal data (wind speed and wind direction) similarities within Jaman North, Jaman South, and Wenchi with their respective production yield. In the Bono region of Ghana, these three areas are crucial for cashew production. The LSTM-DTW-RF model with wind speed and wind direction achieved an R2 score of 0.847 and the LSTM-RF model without these two key features R2 score of (0.74). Both models were evaluated using the augmented Dickey-Fuller (ADF) test, which is commonly used in time series analysis to assess stationarity, where the LSTM-DTW-RF achieved a 90% level of confidence, while LSTM-RF attained an 87.99% level. Among the three municipalities, Jaman South had the highest evaluation scores for the model, with an RMSE of 0.883, an R2 of 0.835, and an MBE of 0.212 when comparing actual and predicted values for Wenchi. In terms of the annual average wind direction, Jaman North recorded (270.5 SW°), Jaman South recorded (274.8 SW°), and Wenchi recorded (272.6 SW°). The DTW similarity distance for the annual average wind speed across these regions fell within specific ranges: Jaman North (±25.72), Jaman South (±25.89), and Wenchi (±26.04). Following the DTW similarity evaluation, Jaman North demonstrated superior performance in wind speed, while Wenchi excelled in wind direction. This underscores the potential efficiency of DTW when incorporated into the analysis of environmental factors affecting crop yields, given its invariant nature. The results obtained can guide further exploration of DTW variations in combination with other machine learning models to predict higher cashew yields. Additionally, these findings emphasize the significance of wind speed and direction in vertical farming, contributing to informed decisions for sustainable agricultural growth and development.

Introduction

Recently, there has been a significant focus on crop yield, which is influenced by various factors such as crop genotype, environment, and management practices (Khaki, Wang & Archontoulis, 2020). Machine learning and deep learning models have been used in different forms to predict crop yield, providing valuable insights throughout the supply chain from pre-production to post-production. In the global economy, one crucial objective of accurately predicting crop yield is to ensure an adequate food supply for nations, including livestock feed and energy resources. This necessitates the development of a crop prediction model that can deliver high-precision results to facilitate effective decision-making.

Machine learning is a set of statistical methods designed to solve specific tasks such as classification or regression by automatically detecting patterns and anomalies in data and making decisions or acquiring skills similar to humans, improving their learning independently over time (Nti et al., 2022; Sagan et al., 2021). Deep learning models, including convolutional neural network (CNN) and long short-term memory (LSTM), have been employed by various authors (Cao et al., 2018; Srivastava et al., 2022; Wang et al., 2022, 2020) to predict crop yield in wheat and other crops. Hybrid deep-learning models have also been studied (Khaki & Wang, 2019) to predict crop yield based on environmental and genotype features. Additionally, machine learning models have been utilized by Ganapathi, Sudarshan & Bhatta (2020), Kalimuthu, Vaishnavi & Kishore (2020), Kumar et al. (2015), Zhou et al. (2022) to predict crop yield.

Over the years, deep learning techniques have been extensively applied to predict crop yield with high accuracy in various crops by authors such as Khaki, Wang & Archontoulis (2020), Khaki & Wang (2019), Ma et al. (2022), Sagan et al. (2021), Tian et al. (2021), Wang et al. (2020) investigated crop yield enhancement in winter wheat using LSTM and remote sensing data. LSTM, a special type of recurrent neural network (RNN), is capable of capturing long-term dependencies (Bhimavarapu, Battineni & Chintalapudi, 2023). It can bridge long time intervals between inputs and analyze temporal patterns at different frequencies, which is advantageous for analyzing crop-growing cycles of varying lengths (Omdena, 2022). Bhimavarapu, Battineni & Chintalapudi (2023) also highlighted that LSTM considers historical values, adjusts itself based on complete patterns, and makes future forecasts. Furthermore, machine learning regression models have proven to be effective for crop yield prediction, as demonstrated by authors such as Keerthana et al., (2021), Panigrahi, Kathala & Sujatha, (2023), Rale et al. (2019), Wang et al. (2016). In fact, Wang et al. (2016) showed that the random forest model produced more accurate estimates in their research. Dynamic time warping (DTW), as defined by Xiao et al. (2023), is an effective method for limited-samples-based crop classification that compares the similarity between two time-series curves, exhibiting reduced sensitivity to training samples. Research according to Bediako-Kyeremeh et al. (2023), Zhao et al. (2021) suggests a gap in the application of DTW in crop yield as most literature discussed DTW in crop mapping (Guan et al., 2016) with limited work on crop yield prediction (Zhao et al., 2021). We aim to integrate DTW into an ensemble of LSTM and Random Forest (RF) models to achieve a higher accuracy model as it has the potential to detect anomalies or aberrations in crop yield data by comparing the temporal patterns with expected norms or historical averages, exhibiting temporal variations due to factors such as weather conditions like wind speed and direction. In the results (Das et al., 2022), the authors realized a novel combination of several deep learning models could not outperform an individual model for weather-based cashew prediction. However, they failed to highlight how important variables such as wind speed and wind direction contributed to the yield prediction, as cashew pruning is essential for yield air circulation is an important aspect of cashew pruning for enhanced yield (Adiga et al., 2020; Balogoun et al., 2015). 1. Can it be demonstrated that DTW can be utilized to assess the similarity of targeted features, such as wind speed (Adiga et al., 2020; Balogoun et al., 2015) and wind direction, in a spatial dataset of cashew crops and produce better predictions than what has been claimed in the literature (Zhao et al., 2021)?

2. Can DTW be integrated into an ensemble learning mechanism to achieve a more accurate model?

To address these questions, we have proposed a learning mechanism framework that combines long short-term memory (LSTM), dynamic time warping (DTW), and Random Forest regressor (RF). The LSTM-DTW-RF approach is expected to provide a more sophisticated and accurate prediction model for crop yield, leveraging the valuable information embedded in wind speed and direction data over time. This integrated methodology enhances the overall robustness and precision of crop yield forecasting in agricultural applications. DTW will enhance spatial analysis within the framework by incorporating specific environmental features wind speed and direction to predict yield in a selected cashew-growing geographical area. The integration of DTW with deep learning models in crop yield prediction is seldom applied but can handle complex temporal dynamics, capture nonlinear patterns, offer flexible sequence alignment (wind speed and wind direction), and enhance feature representation (Zhao et al., 2021; Zuo & Yan, 2018). This approach stands out in its adaptability to the inherent variability in agricultural data and has the potential to provide more accurate and nuanced predictions compared to traditional methods or individual models (Belgiu et al., 2020; Vasilakos, Tsekouras & Kavroudakis, 2022). Adopting a learning mechanism integrated with DTW to assess similarities and differences in instrumental environmental features is essential for predicting crop yield. Moreover, Zhao et al. (2021) applied a combination of DTW-LSTM on MODIS time series data to predict short and long-term winter wheat NDVI.

Materials and Methods

Data

The choice of this dataset was influenced by the environmental conditions required for cultivating cashew crops in a semi-arid area. The data specifically encompassed the three municipalities where cashew cultivation takes place (Balogoun et al., 2015; Okeke & Akarue, 2018) except for fertilizer application data due to unavailability. We utilized a dataset compiled by the Ghana Meteorological Agency (GMet, 2021). This dataset included environmental parameters such as solar radiation, relative humidity, soil drought, and rainfall, collected throughout the entire year from 1999 to 2018, encompassing a span of 20 years.

Cashew yield production data from the Ministry of Food and Agriculture (Ministry of Food and Agriculture of Ghana, 2021) for the municipalities being investigated. The data encompassed the study period of 1999–2018 and were focused on cashew-growing regions, namely Jaman North, Jaman South, and Wenchi. Figure 1 illustrates the geographical positions of our study areas.

Figure 1 Study area of three municipalities (Jaman North, Jaman South, and Wenchi).

Insert map of West Africa and Ghana. Source: ArcGIS. Geographical map showing the location of the three municipalities known for large cashew production in Ghana.

We accessed remote sensing information on soil moisture, wind speed at 2 m, and wind direction at 10 m for the three designated study regions from Prediction of Worldwide Energy Resource Data Access Viewer enhanced (NASA, 2023) spanning the study period of 1999–2018. Overall, the dataset encompasses a total of 517 specific time points in the time series dataset, with nine features. In the preprocessing stage, 67 observations were excluded due to missing values and outliers. A temporal distribution analysis of crop yields over time and wind speed and wind direction over the study period due to the importance of these two environmental variables to our research are visualized in Figs. 2A and 2B. Temporal distribution analysis provides valuable insights into the temporal dynamics of events or phenomena, helping researchers, analysts, and decision-makers better understand and respond to temporal patterns and variations in data. Figure 2A visualizes the cashew yield over the study period for the three study areas in a histogram. A column and line chart for Fig. 2B displays observed trends related to specific periods while identifying anomalies in wind speed and direction distribution. Through an understanding of historical trends, we were able to create predictive models (LSTM-DTW-RF and LSTM-RF) to anticipate future crop yields, assisting farmers in making proactive decisions—an essential goal of this research.

Figure 2 Temporal distribution analysis (A) histogram visualizing trends in crop yield over the study period and (B) line chart visualizing selected environmental variable (wind speed and wind direction) data distribution over the study period.

Model framework

LSTM has shown promising results in crop cultivation, as demonstrated by the study conducted by Das et al. (2022), Wang et al. (2022), Zhao et al. (2021). Additionally, similar positive outcomes have been observed with machine learning regressors, as highlighted by Rale et al. (2019). A prospective avenue involves integrating mapping techniques to enhance crop prediction by considering land use and land cover, a concept explored in the works of Chaves et al. (2021), Feng et al. (2021), Wang et al. (2020), Zhao et al. (2021).

Our proposed model is a fusion of LSTM and RF regression, incorporating DTW. This combined approach is well-suited for time series regression tasks. DTW serves as a valuable tool to measure the similarity between two-time series sequences, particularly when the parameters may have varying lengths or exhibit evidence of time-based warps (Zhao et al., 2021). In Fig. 3, we illustrate the framework of DTW integrated into the ensemble deep learning model.

Figure 3 Deep learning-DTW ensemble model framework.

Model construction

Given a period series observations X1, X2,…, Xn representing input features that are used to build the proposed predictive model. To ensure data quality, missing values were removed using the dropna function and outliers were filtered using the interquartile percentage technique (Q1 = dataset. quantile (0.25) and Q3 = dataset. Quantile (0.75)). For scaling features, we applied the MinMax scaler to numerical features within the dataset and employed one-hot-encoding on categorical features like region or municipality for the random forest, which requires numerical input to process and interpret these variables. The application of both techniques aims to enhance overall compatibility, and model performance while analyzing temporal dependencies and identifying patterns. The dataset underwent an optimal ratio of 70–30% train/test split spanning from 01/01/1999 to 31/12/2018. This split is crucial for interpreting our model’s ability and temporal granularity (Joseph, 2022). The first 70% of chronologically ordered data was used for training, and the remaining 30% for testing, adopting a forward-looking split, where a time series cross-validation was applied, considering the order of temporal data points. This evaluates the model’s ability to make predictions about unseen future data based on past observations. A cross-validation split of 450 data points and an n-split of 7. The number of splits is to strike a balance between reducing the variance in performance estimates compared to fewer folds and maintaining a reasonable computational cost compared to a larger number of folds (Hasan, Kalıpsız & Akyokuş, 2020). However, For the dataset with 450 data points, this resulted in around 315 data points for training and 135 data points for testing. Each split is meant to provide valuable insights into the model’s performance across different segments of our time series data and prevent potential target leakage, we selected target variables, ensuring that necessary information up to the prediction point is used during training and validation. Figure 4 visualizes the time series cross-validation of our cashew environmental dataset.

Figure 4 Cross-validation split chart.

The LSTM model had 50 neurons in the LSTM units and a single dense unit, determined through an iterative process of experimentation and tuning. The activation function used was sigmoid, with recurrent activation set to tanh to capture and process sequential information effectively. The model was compiled using an Adam optimizer with mean square error loss and trained for 20 epochs with a batch size of 32. This parameter tunning choice balanced the need for learning from data with computational efficiency and avoiding overfitting (Rather, 2021). Subsequently, we applied the DTW algorithm separately to compare predicted sequences of individual output wind speed and wind direction at each time step. This allowed us to measure the similarities of these environmental features through feature engineering. DTW distances were calculated between time series in the training and test datasets, and the optimal alignment path was printed. The matrix representation of DTW was computed using the dot product between matrices and specific parameters for window size (2), window type (Sakeochiba), distance metric (FastDTW), step pattern (method = ’symmetricP2), and normalization (len(sequence1) + len (sequence2)).

The model LSTM-DTW-RF equations and DTW algorithm as proposed by Zhao et al. (2021) are expressed below.

LSTM:

Input gate: it=σ(Wiixt+bii+Whiht−1+bhi)

Forget gate: ft=σ(Wifxt+bif+Whfht−1+bhf

Cell gate: ct=ft⊙ct−1+it⊙tanh⁡(Wicxt+bic+Whcht−1+bhc)

Output gate: ot=σ(Wioxt+bio+Whoht−1+bho)

Hidden gate: ht=ot⊙tanh⁡(ct)

where:

xt is the input at time t, σ is the sigmoid activation function, ⊙ denotes element-wise multiplication.

ht−1 is the hidden state from the previous time step.

Wii,Whi,Wif,Whf,Wic,Whc,Wio,Who are weight matrices.

bii,bhi,bif,bhf,bic,bhc,bio,bho are bias vectors.

tanh is the hyperbolic tangent activation function.

DTW:

DTWq(x,y1)=minπ∈A(x,x1)⁡⟨Aπ,Dq(x,x1)⟩1q

where Dq(x,y1) stores distance d(xi,xj1) at the power q.

Algorithm for DTW.

Input: X(t), 0 ≤ t ≤ nT + L is the historical wind speed/direction time series.

T: represents the length of a complete cashew seasonal period.

n: represents the number of seasonal periods.

L: represents the length of the time series of the last incomplete season.

Output: XnT + L + 1, XnT + L + 2, … XnT + L + predicted length.

For i = 0 to predict length-1 do.

A = {WindSpeed_{nT + i}, WindDirection_{nT + i}} //Sequence of wind speed and direction.

For j = 0 to nT-L do.

B_j = {WindSpeed_j, WindDirection_j} //Sequence of wind speed and direction.

C_j = FastDTW(A, B_j) //Calculate DTW distance considering both speed and direction.

End for

K = minindex(C) //Find the index with the minimum DTW distance

X_{nT + L + i + 1} = X_{K + L + i + 1}

End for

Return X_{nT + L + 1}, X_{nT + L + 2}, \ldots, X_{nT + L + \text{predicted length}}

The framework incorporated a random forest regression model to enhance yield prediction accuracy. This was achieved by utilizing DTW similarity scores, along with other input features, as additional features for the model. The input features were a combination of LSTM output and LSTM-DTW output, while the predicted target variable was the output. The random forest model was configured with 100 estimators, max_depth of 5, min_samples_split of 5 and 2, and min_samples_leaf of 3. Random forest feature randomization and regression prediction can be expressed below:

Feature randomization: Let m be the total number of features, and mrand be the number of features considered at each split:

(mrand≤m)

Prediction (Regression): For a regression task, the mean of the individual tree predictions was taken as the final prediction:

y^=1T∑i=1Ty^i

Performance evaluation metrics such as Mean Bias Error (MBE), Root Mean Square Error (RMSE), and coefficient of determination (R2) were used to assess the accuracy of the model. The expressions for these regression metrics are as follows:

MBE=1n∑i=1n⁡(Pi−Oi)

where Oi is the observation value and Pi is the predicted value

RMSE=∑i−1n(yi−y^i)2n

where y^i the predicted value yi is the observed value, n number of a given dataset.

R2=1−∑i=1n⁡(yi−y^i)2∑i=1n⁡(yi−y¯i)2

where y^i is the predicted value and y¯i is the mean value

Results

The precision of the model, considering both wind speed and wind direction through DTW, was assessed using an R2 score of 0.847. In comparison, without DTW, the R2 score stood at 0.74 in Table 1. It also presents statistical information comparing the performance of two models LSTM-DTW-RF and LSTM-RF. Both models were evaluated using the Augmented Dickey-Fuller (ADF) test independently, which is commonly used in time series analysis to assess stationarity (Lina Sjösten, 2022). LSTM-DTW-RF achieved a 90% level of confidence, while LSTM-RF attained an 87.99% level using our estimation with statsmodels in Python. The test statistic measures the strength of evidence against the null hypothesis of a unit root in the data. A more negative test statistic suggests stronger evidence against stationarity. The p-value indicates the probability of observing the test statistic under the null hypothesis. Lower p-values indicate stronger evidence against stationarity (Lina Sjösten, 2022). Figure 5 displays a line chart comparing the model’s performance with and without the independent variables, namely wind speed and wind direction. This provides a complete overview of the model’s performance outcomes with and without these variables and proves predictive power over existing benchmarks.

Table 1 Model results and performance comparison with other benchmarks.

Model	R2	p-value	Level of confidence	
DTW-LSTM (Zhao et al., 2021)	0.825			
LSTM-RF (our model)	0.74	1.3153912e−01	87.99%	
LSTM-DTW-RF (our model)	0.847	1.4028797e−07	90%	
Notes:

A higher R2 score, closer to 1.0, indicates that a larger proportion of the variability in the target variable can be explained by the model.

The p-value indicates the probability of observing the test statistic under the null hypothesis. Lower p-values indicate stronger evidence against stationarity.

Figure 5 Line chart comparing the model’s performance with and without wind speed and wind direction using respective R2 score.

Notes: w–wind speed, wd–wind direction.

Furthermore, Table 2 presents metric scores reflecting the comprehensive performance of the model, including the individual performance metrics for each municipality. The various metrics for three municipalities were Jaman North (MBE = 0.231, RMSE = 0.802, R2 = 0.742), Jaman South (MBE = 0.22, RMSE = 0.883, R2 = 0.835), and Wenchi (MBE = 0.212, RMSE = 0.746, R2 = 0.702). Figures 6A–6C showcase a line regression chart for RMSE, R2, and MBE. These charts utilize evaluation metrics to visually illustrate the correlation between the predicted target variable (each municipal wind speed, wind direction) and the target variable (municipal production) in the dataset. This is crucial as production is primarily influenced by environmental factors, particularly wind speed and wind direction. The blue, red, and black “dot” symbols in the chart represent each municipality’s computation of metrics, including mean bias error (MBE), root mean square error (RMSE), and R-squared (R2), wind speed, wind direction, and production.

Table 2 Individual municipality model performance.

Municipalities	MBE	RMSE	R2	
Jaman North
Jaman South
Wenchi	0.231
0.22
0.212	0.802
0.883
0.746	0.742
0.835
0.702	
Note:

R2 evaluates the accuracy or goodness of fit of the model predictions; MBE whether it tends to systematically overestimate or underestimate the target variable; RMSE indicate better model performance.

Figure 6 Individual metric score regression line chart (RMSE, MBE and R2).

The trend line with a slope of 1:1, indicating a direct correspondence between predicted and actual values; blue, red, and black “dot” symbols on the chart represent each municipality’s computation of metrics, MBE, RMSE, and R2, wind speed and wind direction; Y-axis of the chart represents the average forecast values of these metrics for each municipality, providing a visual representation of a balanced and aggregated view of the model’s predictions for each municipality; X-axis is dedicated to discrete values associated with the target variable, which, in this instance, pertains to the production levels of each municipality.

The Y-axis of the chart represents the average forecast values of these metrics for each municipality’s wind speed, and wind direction, providing a visual representation of a balanced, aggregated, and facilitating a comparative analysis view of the model’s predictions for each municipality. Choosing this alternative proved superior as opposed to depending on a solitary prediction, average forecast values are frequently computed by averaging or taking the mean of multiple predictions (Shang & Booth, 2020). This approach proved effective in diminishing the influence of outliers or discrepancies in individual predictions, resulting in a more reliable and representative estimate of the anticipated outcome. Conversely, the X-axis is dedicated to discrete values associated with the target variable, which, in this instance, pertains to the production levels of each municipality (Divecha, Tullu & Karande, 2023). By illustrating the correlation between predicted and actual values, the chart facilitates comparative analysis, providing insights into the model’s performance across the three municipalities. The 1:1 slope trend line serves as a benchmark for assessing the accuracy of the predictions.

The DTW similarity distances for the annual average wind direction were recorded as 270.5 SW for Jaman North, 274.8 SW for Jaman South, and 272.6 SW for Wenchi. In terms of the annual average wind speed, the DTW similarity distances varied within the range of ±25.72 for Jaman North, ±25.89 for Jaman South, and ±26.04 for Wenchi. The performance of the DTW model regarding wind speed (measured in km/h) and wind direction (measured in X°) across the three municipalities throughout the twenty-year study duration is visually depicted in Figs. 7A and 7B. These graphical representations offer insights into the model’s effectiveness in capturing and comparing wind characteristics across different locations and periods. The inclusion of DTW significantly contributed to the enhanced accuracy of our results. The metrics table indicates that Jaman South performed the best among the cashew-growing areas, excelling in wind speed determination, while Wenchi ranked second and performed better in wind direction. The geographical location of Wenchi on the western side of Jaman North and South may explain its higher wind direction. Jaman South had a high wind speed, which contributed to its high-yield production which promotes sustainable farming (Beacham, Vickers & Monaghan, 2019; Kalantari et al., 2018; van Delden et al., 2021). The DTW model effectively analyzed the similarity of spatial data, demonstrating its effectiveness for spatiotemporal analysis.

Figure 7 DTW similarity distance line chart for municiaplities.

(A) Wind speed; (B) wind direction. ±; range, SW—southwest winds, km/h—kilometers per hour X°—degrees.

Discussion

We evaluated our model using metrics such as MBE, RMSE, and R2 for the three (3) cashew crop-growing areas over the study period of 1999 to 2018. In assessing the reliability and generalization of our findings, our model exhibited outstanding performance on the training data but struggled with new, unseen data, indicating overfitting during the municipality modeling. This occurred because our model was memorizing the training data and capturing noise instead of discerning underlying patterns.

To address this issue of performance discrepancies, time series cross-validation was implemented during model development using a 7-fold data split, a commonly used approach (Hasan, Kalıpsız & Akyokuş, 2020). Conducting cross-validation mitigates overfitting and assesses the impact of features on model performance, thorough feature engineering, by creating time-based features such as day of the week, month, and year in the data, aimed to capture temporal patterns, aligning with our outlook and yielding the desired results and offering a robust estimate of its generalization performance. Furthermore, another provision to tackle discrepancies, is the construction of our proposed ensemble model (LSTM-DTW-RF), as suggested by Zhao et al. (2021), the utilization of this ensemble approach played a crucial role in mitigating the influence of individual model biases, resulting in an overall improvement in our model’s performance and the generation of more accurate findings presented.

In the process of feature selection and model ablation, concerns arose about the selection of wind speed and wind direction as features, Adiga et al. (2020) and Balogoun et al. (2015) factored wind speed as a necessary environmental factor that could be considered for cashew yield. However, the relationship between wind speed, wind direction, and crop yield is an important factor in agriculture. Wind speed and direction can influence various aspects of crop growth, including pollination and transpiration. Wind speed and direction can help regulate temperature fluctuations in the crop environment, preventing heat stress during hot days and reducing frost damage during cold periods, ultimately affecting crop yield (Gobin, 2018). Based on Adiga et al. (2020), Balogoun et al. (2015), and Gobin (2018), the assumption that these features might effectively represent the underlying patterns in the data, can potentially be relevant. In corroborating this fact, using the LSTM-RF model, the LSTM handled non-stationarity as it is assumed that the underlying data-generating process exhibits some level of stationarity (Markovic et al., 2023). In the LSTM-DTW-RF model, the DTW handled non-stationarity through warping wind speed and direction, as it is beneficial to preprocessing the data to achieve stationarity (Zhao et al., 2021). Also, The RF model provided feature importance that aided in model interpretability, LSTM models are often considered less interpretable due to their complex architecture, therefore, it was appropriate to use RF in the prediction and extracting meaningful insights for agricultural decision-making. Overall, both models demonstrate adequate evidence against the presence of a unit root in the data, indicating stationarity as indicated in the presentation of the results.

DTW enhances crop yield prediction, as it can offer valuable insights and augment the analysis of temporal patterns in yield data, potentially leading to improved decision-making in agriculture. The outcomes presented affirm the effectiveness of integrating DTW and considering both wind speed and wind direction, a consideration overlooked by other studies. While benchmark results (Zhao et al., 2021), incorporated DTW with a deep learning model and utilized NDVI, demonstrated good accuracy, our model surpassed this by incorporating crucial variables essential for thriving cashew growth, such as wind speed and wind direction. Although (Das et al., 2022), proposed individual model predictive accuracy over a combined model, we are confident a model complexity such as the LSTM-DTW-RF model which involves multiple components with different characteristics and interaction, data characteristics, experimental design proffers a higher predictive accuracy.

While previous studies have used TWDTW as a standalone model for classification, our research suggests that using DTW with the right ensemble deep learning models can yield better results when identifying key environmental parameters for crop yield. Wang et al. (2020) focused on using the LSTM model with Modis LAI products and the time-weighted dynamic time warping (TWDTW) variant of DTW to predict the yield of winter wheat in Henan Province, China, but did not consider the effects of wind speed and wind direction on yield. Similarly, Chaves et al. (2021), Wang et al. (2022), and Zhao et al. (2021) discussed the role of TWDTW in determining the area of the crop but overlooked the impact of wind speed and wind direction.

Our results align with the findings of Zhao et al. (2021); however, they applied LSTM-DTW and DTW to predict NDVI winter wheat where their best model performance attained an R2 score of (0.825), while their model did not consider wind speed and wind direction our model factored in wind speed and wind direction as an independent variable using LSTM-DTW-RF and LSTM-RF which showed high yields in Jaman South, Jaman North, and Wenchi, and highlighting the influence of wind speed and direction in the under-studied municipalities. Balogoun et al. (2015) applied wind speed in their environmental parameters to predict cashew yield, however, they applied baseline methods in their prediction. The precision of deep learning ensembled predictions outperforms baseline model predictions (Khaki, Wang & Archontoulis, 2020). Unlike Chaves et al. (2021), who only considered the harvest period, we analyzed the entire year, including the flowering and fruit development stages of the cashew crop, as well as the impact of wind speed and direction during the harvest period on yield.

It is noteworthy that this improvement is particularly relevant given the semi-arid conditions in which cashew crops thrive (Adeigbe et al., 2015; Balogoun et al., 2015) while the findings of our study may provide valuable insights into cashew crop cultivation over the study area, conducting validation studies in diverse geographical contexts and considering local-specific factors are essential steps to assess the applicability of the findings beyond the study region. However, with some cashew growing areas close to the study areas such as Cote d’Ivoire which shares an international boundary with the area under study, this model is expected to generalize beyond its geographical area due to similar socio-economic factors, agricultural practices, relatable environmental factors and market dynamics (UNCTAD, 2021). Our LSTM-DTW-RF model will contribute to advancing knowledge in crop yield prediction by providing a more sophisticated, accurate, and nuanced approach to understanding the impact of environmental factors on agricultural outcomes. This contributes to the ongoing efforts to enhance the precision and reliability of crop yield forecasting, which is vital for sustainable and efficient agricultural practices.

Implementing our proposed model in practical agricultural scenarios, it’s crucial to acknowledge that state departments, including the Ghana Meteorological Agency (GMet), Ministry of Food and Agriculture (MoFA), and NASA Prediction of Worldwide Energy Resource Data Access Viewer enhanced (POWER|DAVe) play a pivotal role in ensuring the availability of high-quality, reliable, and pertinent data (encompassing weather patterns, soil conditions, crop types, and historical yields). These data form a foundational element for effective model development. Simultaneously, considerations should extend to the model’s scalability, exploring computing power options and infrastructure such as cloud computing while optimizing algorithms for heightened efficiency and performance.

Moreover, for practical applications and informed decision-making based on research findings, it is imperative to actively engage end-users in the development process. This involvement allows for a better understanding of their needs and preferences, facilitating the creation of user-friendly interfaces and dashboards. The outputs of the model must remain interpretable (Jain & Choudhary, 2022), providing clear guidance for practical decision-making. Additionally, comprehensive documentation detailing the model’s functionality and the variables influencing predictions is essential (Das et al., 2022). These considerations extend beyond technical aspects to encompass broader concerns such as data privacy and security measures (Al-Adhaileh & Aldhyani, 2022), adaptability to local conditions (Balogoun et al., 2015; UNCTAD, 2021), meaningful stakeholder engagement (Evans, Mariwah & Antwi, 2014), adherence to regulatory compliance, and conducting efficient cost-benefit analyses, all of which can help reduce barriers to the adoption of this model. While cashew farmers in the study area currently lack adequate records of fertilizer usage (SNRD, 2019; UNCTAD, 2021), addressing this limitation through accurate predictions from the proposed and proper fertilizer application along with record-keeping would enhance decision-making. With a period of fertilizer application history model algorithm can be upgraded to factor in the fertilizer application feature to make the prediction more rounded.

Conclusions

This study underscores the economic significance of cashew cultivation and highlights the importance of considering environmental factors, particularly wind speed and direction, in predicting crop yields. By employing advanced deep learning techniques like LSTM and random forest models alongside dynamic time warping (DTW), researchers achieved notable improvements in yield prediction accuracy in Ghana’s Bono region. Notably, the LSTM-DTW-RF ensemble model, incorporating wind-related features, outperformed models lacking these variables, indicating the critical role of environmental data in predictive modeling. Regional variations in wind patterns further underscored the necessity of tailoring predictive models to specific environmental conditions. This study’s findings not only demonstrate the efficacy of DTW in capturing spatiotemporal data similarities but also emphasize the potential for informed decision-making in agricultural practices contributing to sustainable agricultural growth and development efforts. Further research could delve into refining DTW variations and exploring additional machine learning techniques to enhance predictive capabilities for optimizing agricultural productivity in cashew-producing regions and beyond.

Supplemental Information

Supplemental Information 1 Supplementary Tables.

Additional Information and Declarations

Competing Interests

Author Contributions

Data Availability

The authors declare that they have no competing interests.

Bright Bediako-Kyeremeh conceived and designed the experiments, performed the experiments, analyzed the data, prepared figures and/or tables, authored or reviewed drafts of the article, and approved the final draft.

TingHuai Ma conceived and designed the experiments, performed the experiments, analyzed the data, authored or reviewed drafts of the article, supervision, and approved the final draft.

Huan Rong performed the experiments, analyzed the data, authored or reviewed drafts of the article, co-Supervision, and approved the final draft.

Benjamin Kwapong Osibo conceived and designed the experiments, performed the experiments, analyzed the data, prepared figures and/or tables, authored or reviewed drafts of the article, and approved the final draft.

Lorenzo Mamelona conceived and designed the experiments, performed the experiments, analyzed the data, prepared figures and/or tables, authored or reviewed drafts of the article, and approved the final draft.

Isaac Kofi Nti conceived and designed the experiments, performed the experiments, analyzed the data, authored or reviewed drafts of the article, and approved the final draft.

Lord Amoah performed the experiments, analyzed the data, prepared figures and/or tables, authored or reviewed drafts of the article, and approved the final draft.

The following information was supplied regarding data availability:

The data is available at figshare: Bediako-Kyeremeh, Bright (2023). Cashew Env. Dataset.csv. figshare. Dataset. https://doi.org/10.6084/m9.figshare.23650551.v1.

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
