# Peer review of "Effects of wind speed and wind direction on crop yield forecasting using dynamic time warping and an ensembled learning model"

_PeerJ, doi:10.7717/peerj.16538_

## Round 0.1 · original submission · Major Revisions

Please incorporate all comments raised by reviewers and please submit the revision along with a point-to-point rebuttal letter. Please also improve the language and the graphical presentation of the paper.

**Language Note:** The Academic Editor has identified that the English language must be improved. PeerJ can provide language editing services - please contact us at copyediting@peerj.com for pricing (be sure to provide your manuscript number and title). Alternatively, you should make your own arrangements to improve the language quality and provide details in your response letter. – PeerJ Staff

·

Basic reporting

Comments
Topic: It is well-formed, covers the work, and is intriguing.
Abstract: It is all around (Introduction, methods, results, hypothesis, and conclusion)
Lines: 23-28: Long sentence breakdown also RMSE, MBE, and R2 scores of (x) and (y) must be added. Line 29 The coefficient (20) determinant (of).
Line 46: define (TWDTW)
Line 51: define POWER|DAVe and MODIS in full if it is an abbreviation.
Line 79: (Xiao et al., 2023) authors defined Dynamic Time Warping (DTW) and cite the beginning paragraph correctly. Xiao et al., (2023)
Line 89: I believe the authors meant ‘crop yield prediction’ and not ‘crop yield’
Line 197: Our proposed, lowercase ‘o’
Line 221: Cite this Wang et al., 2022) authors properly, -Wang et al., (2022) authors
Figure 3: Remove the black line under the image
Figure 4: Enhance Figure 4.- Colors should be brighter/Outstanding. Add Legend X and Y explained
Figure 5: Write in full (Obs) for all matrix results charts and adjust contrast to look sharper
Figure 6: Position the Chart vertically and expand each to be clearer rather than side-by-side and moved to the result section
Elaborate further model before model evaluation. (RMSE, MBE, and R2) and also define the character “x” in charts and repetition of cited dataset sources.
Line 165-171: The first paragraph should be moved to the discussion section
Discussion
Lines 202-206: long sentence therefore breakdown
Lines 227-228: moved to the results section
Lines 229-251: should be fused and summarized, I see a repetition of discussed results within those lines.
Line 256: use “could be used” to replace “can be used”
Lines 257-258; reframe sentence.
Line 263-266: sentence too long breakdown
Duplicated reference (lines 256-280 and Lines299-301)
Summary
This research work has been reviewed and seems to have some novelty, integrating a warping variant in a deep-learning model using environmental variables, and empathizing targeted variables to determine its effects on crop yield.
The topic “Effects of wind speed and wind direction on crop yield forecasting using dynamic time warping and an ensembled model” is well-couched, it covers the entire work and is meaningful. The abstract is concise and entails the structure of an abstract. The introduction articulated and captured the problem and cited references to previous works and framed the research question to correspond to the problem. Within the introduction current works were also cited which cover all three models integrated.
The materials and methods section, commencing from datasets was indicated and interpreted, it shows that the dataset was multi-sourced and cited all sources in the work. In the methods, the authors need to elaborate further on the model’s dynamics and techniques deployed before the evaluation using the matrix. Matrix mathematical expressions were stated, and variables were interpreted. Model architecture image, explained in steps with a dynamic warping algorithm provided. Although the model steps are stated, authors should state the techniques such as (data splits, number epochs, etc.) applied to work to achieve results.
With regards to results, I suggest the few lines stated above seem more of a discussion point than a result and therefore should be moved accordingly. Charts match results and just need a little image enhancement.
The discussion cited related works’ results and models and juxtaposed them with the author’s work, stating their hypothesis, therefore, stating how their work differs and is more advanced than others. A few repetitions in the lines stated above, therefore, I suggest those lines are fused and well structured. The Conclusion, lines stated needs to be framed grammatically and delete repeated references.
Overall, the work has some high level of novelty, the topic and abstract are tactically drafted, the scientific methods applied are satisfactory and the results are articulated and discussed appropriately. The conclusion stated achievements from the work and methods used and future works proposed. I suggest this work be accepted after minor corrections are done.

Experimental design

no

Validity of the findings

no

Additional comments

no

Reviewer 2 ·

Basic reporting

The topic and abstract are a well-structured and comprehensive piece of work. It highlights that the content covers all necessary components, such as the introduction, methodology or procedures, results, discussion fused with the research premise, and conclusion which can capture the reader's interest. The work gives another dimension of features with feasible effects of crops especially in tropical crops. The authors made good use of current references as 90% of cited works are within the last ten years. Model construction is satisfactory, and the main model (DTW) algorithm is stated in the text. Some comments are as follows:
1. For the Abstract: Lines: 23-28: has a long sentence that can be broken down also RMSE, MBE, and R2 as read but without respective result scores and must be indicated in the abstract as shown in the table under the results section for first-hand information by the journal readership. In line 29 The coefficient of determinant typo error must be addressed.
2. For the Introduction: In line 46 and line 51, define the abbreviations of the following in the write-up TWDTW, POWER|DAVe, and MODIS for readers to understand better. Also, in lines 165-171 and lines 227-228, I have a strong conviction are discussion points, therefore move them to section (Discussion).
3. For the References: Lines 256-280 and lines 298-301 has repeated reference. Please address it.

Experimental design

No comment

Validity of the findings

1. In figure 3 the black line/bar underneath the image must go off and in figure 1, geographical coordinates are small therefore use a bigger text size. In figure 4 and figure 5 image contrast enhancement (either colors or image file type is not making it stand out well)
2. In lines 229-251, which should be fused and summarized, I see a repetition of discussed results within those lines, while combining in lines 165-172, respectively.

---

## Round 0.2 · Minor Revisions

Please incorporate the following comments and submit a point-to-point rebuttal letter.

This is an interesting topic but there are a number of points that need clarification.

I believe that the goal is to assess the importance of wind speed and wind direction on crop yield forecasting. If so, shouldn't predictions with and without this variable included be presented, so that we can evaluate the importance of these variables? Or is the goal only to be modeling (predicting) wind speed and wind direction? If the latter, then the title is misleading and the goal should be made clearer in the abstract and paper.

Doesn't Figure 3 suggest very little predictive power of the model? Generally, this type of data should be presented in a scatter plot. Also, is Figure 3 for the test set? please specify. I don't understand how R2 values of 0.7-0.8 are being found given Figure 3.

I don't understand Figure 4. We are shown model vs observed. I don't understand how one calculates RMSE, etc, separately for the observed values and the predicted values.

I don't understand how Figure 5 shows the "performance of the DTW model". What is Figure 5 trying to show?"

·

Basic reporting

Accepted for publication. The author has diligently addressed the suggested comments, resulting in an improved manuscript.

Experimental design

no

Validity of the findings

no

Additional comments

no

Reviewer 2 ·

Basic reporting

no comment

Experimental design

no comment

Validity of the findings

no comment

---

## Round 0.3 · Major Revisions

The new reviewers, who are experts in machine learning, have given comments on your paper. Please incorporate these thoroughly and also submit a point-to-point rebuttal letter.

·

Basic reporting

accepted for publication

Experimental design

no

Validity of the findings

no

Additional comments

no

Reviewer 2 ·

Basic reporting

no comment

Experimental design

no comment

Validity of the findings

no comment

Additional comments

no comment

Reviewer 3 ·

Basic reporting

Sharing the code repository (and/or the data, if possible), although not a mandatory requirement, would significantly enhance the paper's value by facilitating reproducibility and further research. It will also clear out some of the methodology related questions, based on the paper itself, which I have personally had.

Experimental design

Data Section: Enhancing this section with more comprehensive statistics about the dataset—including its size, the number of excluded observations, and the distribution of the variables — would greatly benefit the reader.

Methods: Considering the time series nature of the data, clarification on how the 70%-30% train/test split was implemented would be valuable. Specifically, it's important to specify if the predictions were made in a forward-looking manner (as opposed to backward-looking), and to address the potential for target leakage. Additionally, providing information on the size of the data would aid in understanding the model's generalizability.

More in-depth technical details regarding the model's implementation, parameter tuning, and the reasoning behind certain model configurations, such as the choice of 20 training epochs, would significantly improve the reader's comprehension.

Furthermore, there seems to be ambiguity in the use of the term 'target', which is referred to in the context of both the model output and other features. For example, the statement 'for the target variable transformation scaling, we used a one-hot-encoding technique on the categorical feature and production' suggests a possible confusion. Given that a random forest regressor (not a classifier) is employed, it implies that these categorical features are not the output targets, according to my interpretation.

Validity of the findings

While the paper presents promising results in crop yield prediction accuracy, enhancing the clarity of results reporting could further strengthen the study.

Figure 2: Providing details on the dimensionality of the inputs and outputs would greatly enhance the reader’s comprehension.

Figure 4: Despite reviewing the authors' response to the previous reviewer, the labeling of the figure axes remains unclear to me. Typically, labels such as 'Observed' and 'Model' suggest a comparison between the target variable and the predicted target variable, with a 45-degree line indicating good model performance.

However, in this context, it appears that the three points in each figure represent three different municipalities, with their respective metrics plotted on the y-axis. If so, this aspect is not immediately clear. Furthermore, the purpose of the red lines in each graph is ambiguous, and the nature of the x-axis (whether it is meant to be discrete) is not evident.

Clarifications on these points would significantly aid in interpreting the figures and understanding the model’s performance across the municipalities.

Reviewer 4 ·

Basic reporting

Rationale and Novelty:

1. What specific gap in the existing literature does this study aim to address, and how does it contribute to advancing knowledge in the field of crop yield prediction?

2. What makes the integration of Dynamic Time Warping (DTW) with deep learning models novel or unique compared to previous approaches in crop yield prediction?
Methodological Rigor:

3. How were the environmental variables selected for inclusion in the dataset, and were any potentially influential variables overlooked?

4. Can you provide more detail on the process of model validation, including any cross-validation techniques utilized to assess the generalizability of the results?

5. Were there any assumptions made during the construction of the model or the interpretation of results that might impact the validity or reliability of the findings?
Model Performance and Interpretation:

6. While the study reports high accuracy metrics for the proposed model, how do these results compare to existing benchmarks or alternative methodologies?

7. Can you discuss any instances where the model performance deviated significantly from expectations, and how were these discrepancies addressed or explained?

8. How confident are you in attributing the observed differences in crop yield prediction accuracy to the incorporation of DTW into the model, as opposed to other factors or variables?

Experimental design

9. To what extent do you believe the findings of this study are generalizable beyond the specific geographical and temporal contexts of cashew crop cultivation in the selected regions?

10. How feasible would it be to implement the proposed model in real-world agricultural settings, considering factors such as data availability, computational resources, and practical usability?

Validity of the findings

11. Have you considered potential limitations or barriers to adoption that might arise when translating the research findings into practical applications or decision-making processes?

Additional comments

Could you please provide more details on the comparison of your model's performance against existing benchmarks? Additionally, how were discrepancies in model performance addressed, and what implications do these have for the validity of your findings?

---

## Round 0.4 · Minor Revisions

Please incorporate the final minor revisions and submit the paper with response letter.

Reviewer 2 ·

Basic reporting

no comment

Experimental design

no comment

Validity of the findings

no comment

Additional comments

no comment

Reviewer 3 ·

Basic reporting

No comment

Experimental design

No comment

Validity of the findings

The issues I previously raised have been thoroughly addressed by the authors, and I appreciate your efforts. I have two additional comments:

1. Please correct the misprint in Figure 6B (Wenchi), which appears to be duplicated instead of showing Figure 6C
2. It is advisable to apply the min-max scaler using only the minimum and maximum values from the training set, rather than from the entire dataset, as indicated in the code

I do not think these comments necessitate another round of review.

Reviewer 4 ·

Basic reporting

No comment

Experimental design

No comment

Validity of the findings

No comment

Additional comments

No comment

---

## Round 0.5 · accepted · Accept

All comments are duly incorporated. So, the paper is accepted.